# OpenReview forum: "How to Catch an AI Liar: Lie Detection in Black-Box LLMs by Asking Unrelated Questions"
_ICLR.cc/2024/Conference — ICLR 2024 poster_

### Official Review · Reviewer_cT55 · 2023-10-22

**Soundness:** 3 good
**Presentation:** 2 fair
**Contribution:** 3 good
**Rating:** 8
**Confidence:** 4

**Summary:**

The authors propose a simple methodology to detect whether an LLM’s response is a lie (an answer that is false, even though the model can produce the correct answer). Authors use a set of fixed elicitation questions to probe a model and only use the responses to these followup questions to predict whether an LLM lied. The results demonstrate that this methodology is surprisingly accurate and has intriguing generalization properties.

**Strengths:**

1. The problem definition is strong and seems impactful. That is, the concept of lying for an LLM is formalized as LLMs producing a false statement while being able to accurately produce the correct statement. The toolbox proposed in the paper contributes to having practical reliability safeguards for existing LLMs.

2. I am surprised and excited to see that the lie detector generalizes to different architectures and distributions of input text, which further adds to the value of the proposed methodology. Specifically, this allows the design of general-purpose lie detectors that are cheap to implement across a variety of settings.

3. I am also very curious about the underpinnings of the observed behavior, which is very novel and not have been discussed before to the best of my knowledge. Specifically -- what are the mechanisms that relate the lying behavior to the answers to seemingly unrelated questions? Although the paper does not provide concrete explanations as to why the classifiers surprisingly work very well, these properties motivate several interesting questions to pursue for future work.

4. The authors did extensive evaluations to make sure that the results were not due to spurious correlations, such as simply picking up on the repeated lying behavior, but have more going on (i.e. the fact that responses to ambiguous questions can also predict the lying behavior).

**Weaknesses:**

1. (Lack of Descriptive Discussion) I would have loved to see more on what the classifier may be picking on; or what the suggested inductive biases may be. As it stands, it is rather a mysterious result. However, I do not find this to be a significant limitation, as the results are thorough, and future work can further investigate this phenomenon.

2. (Labeling for Correctness) The way authors labeled an answer as correct/incorrect appeared slightly brittle to me (in that the correctness of a model’s response is defined as whether an LLM’s response contains the ground truth entity). For instance, a response could still mention the ground truth entity and be wrong. However, I acknowledge that evaluating natural language generations is challenging and still an open research area, thus this limitation is hard to address.

3. (Lack of ablation) I would also be curious to see how redundant the set of 48 questions is, the set of questions and the number of questions is somewhat arbitrary and I’m curious how the authors reached this number. Can we achieve the same performance with only 2 ambiguous elicitation questions? What happens if the authors just keep the top 2 questions with the largest weights? What does that imply?

4. (Presentation) There is a lot to unpack, and I had a bit of a hard time following Sections 4 and 5. I think one factor was that the presentation of the dataset is rather scattered (4.1, 4.2), and the methodology is rather scattered (4.3, 5.2). There are also quite a bit of details deferred to the Appendix (e.g. finetuning details), which makes reading very nonlinear.


Minor

1. There are a lot of numerical data in the paper that are inline, which makes it a bit hard to see when one needs to recall previous related results. I’d appreciate having most of the results contained in tables in the main text, however, I can also see that the space is a limiting factor.

**Questions:**

1. Regarding W3: Have the authors performed any ablations for the number of questions? Otherwise, how did the authors reach the number 48 / other more granular numbers for each of the factual / ambiguous questions?

2. Similar to Q1: I’d be curious to see a study around performing lasso/feature selection experiment to see how far this number can be reduced or is saturated.

---

> ### Author Response · Authors · 2023-11-16
> **Author Response**
>
> Thank you very much for your review of our work, we appreciate your comments and are glad that you found the paper both surprising and exciting!
>
> > (Labeling for Correctness) The way authors labeled an answer as correct/incorrect appeared slightly brittle to me (in that the correctness of a model’s response is defined as whether an LLM’s response contains the ground truth entity). For instance, a response could still mention the ground truth entity and be wrong. However, I acknowledge that evaluating natural language generations is challenging and still an open research area, thus this limitation is hard to address.
>
> We agree that this can definitely be a challenging area to assess and so to mitigate this we did make sure to randomly check a small number of the examples by hand in order to make sure nothing was getting too obviously misclassified. Our approach appeared to work well. We also made sure to think through failure modes and avoid them, e.g., we made sure to look for the full word, not just potential substrings (e.g., the word “no” instead of the substring, which can be found in words such as “kNOwledge”). Additionally, we had alternative evaluations methods for e.g. the translation questions where we asked a separate model. Finally, the questions we ask are mostly relatively clear and simple, and the LLMs usually answer in a simple and straight-forward way. The evaluation method is thus less likely to fall prey to the particular failure mode you describe.
>
> > (Presentation) There is a lot to unpack, and I had a bit of a hard time following Sections 4 and 5. I think one factor was that the presentation of the dataset is rather scattered (4.1, 4.2), and the methodology is rather scattered (4.3, 5.2). There are also quite a bit of details deferred to the Appendix (e.g. fine-tuning details), which makes reading very nonlinear.
>
> Thank you for highlighting these points - in the revised paper that we have uploaded we have slightly restructured Sections 4 and 5 in order to make the ideas easier to follow. We have collated the methodology by moving most of Section 5.2 into a new Section 4.
>
> > (Lack of ablation) I would also be curious to see how redundant the set of 48 questions is, the set of questions and the number of questions is somewhat arbitrary and I’m curious how the authors reached this number. Can we achieve the same performance with only 2 ambiguous elicitation questions? What happens if the authors just keep the top 2 questions with the largest weights? What does that imply?
> > Regarding W3: Have the authors performed any ablations for the number of questions? Otherwise, how did the authors reach the number 48 / other more granular numbers for each of the factual / ambiguous questions?
> > Similar to Q1: I’d be curious to see a study around performing lasso/feature selection experiment to see how far this number can be reduced or is saturated.
>
> Thank you for this interesting suggestion for further exploring the properties of our lie detection method. In order to explore this question we have added an **additional experiment** in Appendix D.7 (that we shall also detail here) regarding an ablation where we vary the number of questions (features) used for our lie detector on the in-distribution examples. The results can be seen in Figure 11 (also available at [this link](https://imgur.com/a/LPBEHLI)) and show the AUC for both the Binary and Logprobs detector as on the x-axis the number of features is increased. The selection method for additional features was performed using scikit-learn’s $\texttt{SequentialFeatureSelector}$ which greedily picks the best feature to add to the existing set according to cross-validation performance. The selection was performed on the train dataset for our lie detector (consisting of lies and truths generated by GPT-3.5 with direct and CoT prompts). We then compute the AUC of the detector as the number of features is increased, both on the training distribution and on two of the generalization studies (those on Alpaca-7B and Vicuna-7B, for which the detector and the feature choice are frozen to those obtained from the training distribution). Figure 11 shows how, while high in-distribution AUC can be obtained with a few features (particularly for some of the elicitation question groups), increasing the number of features generally leads to better performance for out-of-distribution cases, although the trend is not monotonous. This is particularly true for the binary detector.

---

> > ### Comment · Reviewer_cT55 · 2023-11-17
> > **Response to rebuttal**
> >
> > Thank you for your rebuttal! I appreciate the presentation changes and the additional experiment.
> >
> > **Additional Experiment:** This seems pretty interesting and surprising -- am I correctly reading that having only 1-2 questions can result in 90%+ AUC in certain settings (i.e. mostly for 'in-distribution', and certain cases for generalization experiments)? How do you interpret this?
> >
> > Also, this makes me wonder whether you use not only the in-distribution model but a second model to pick questions and observe the performance of the third model. Do you think the questions picked this way could generalize better, perhaps?
> >
> > Minor: The caption for Figure 11 and the subcaption for Figure 11a has parentheses that are left open.

---

> > > ### Author Response · Authors · 2023-11-20
> > >
> > > Thank you for these additional insightful comments.
> > >
> > > > This seems pretty interesting and surprising -- am I correctly reading that having only 1-2 questions can result in 90%+ AUC in certain settings (i.e. mostly for 'in-distribution', and certain cases for generalization experiments)? How do you interpret this?
> > >
> > > The reviewer is correct in their interpretation. However, we would like to stress that, while 1-2 questions lead to high performance for *some* generalization experiments, one is interested in having the detector as broadly applicable as possible to new cases, where one may not have labelled ground truth. As such, the fact that having more questions generally leads to better performance for the generalization studies presented in Figure 11 suggests that one has to aim to use as many questions as possible, even if a few of them may be needed in specific cases.
> > >
> > > Moreover, we would like to highlight how such a high AUC with few questions is mostly obtained when using logprobs, rather than the binary detector. The benefit of more questions is therefore more pronounced when a user does not have access to logprobs, but can only rely on binary answers.
> > >
> > >
> > > > Also, this makes me wonder whether you use not only the in-distribution model but a second model to pick questions and observe the performance of the third model. Do you think the questions picked this way could generalize better, perhaps?
> > >
> > > If we correctly understand the reviewer's suggestion, they propose to select the best features (for instance within `SequentialFeatureSelection`) by training the detector on a dataset and using the performance on another one, in contrast to using cross-validation in-distribution performance. This is an interesting suggestion and could be implemented in practice. We believe this can counter some dataset-specific bias and may thus generally improve the performance of the reduced set of questions on a novel dataset. However, it seems likely that, for all new datasets on which one may want to apply the trained detector, there are datasets for which a different choice of features would be more beneficial. As such, while the reviewer's proposal may help to reach better performance with few features ``on average'' over the set of possible datasets, it would not change the fact that, to ensure broad applicability, more questions are better, as we cannot fundamentally know which questions will be more useful for novel datasets.
> > >
> > > > Minor: The caption for Figure 11 and the subcaption for Figure 11a has parentheses that are left open.
> > >
> > > Thank you for pointing out this issue, which we now fixed. We also realized that the legend in panels b and c were incorrect and we now fixed that too and uploaded a new paper version; you can find the new figure [here](https://imgur.com/a/aSGFe9f). The overall conclusions are the same, so the preceding arguments stay valid.

---

> > > > ### Comment · Reviewer_cT55 · 2023-11-21
> > > > **Response to authors**
> > > >
> > > > Acknowledging your response, thanks for the clarifications and updates.

---

### Official Review · Reviewer_eJWS · 2023-10-30

**Soundness:** 3 good
**Presentation:** 1 poor
**Contribution:** 2 fair
**Rating:** 5
**Confidence:** 3

**Summary:**

The paper proposes a lie detector for LLMs that works by answering unrelated follow-up questions and requires no access to internals. This is an interesting topic that relates to hallucinations but still has not received a lot of attention on its own. The author's method strikes through simplicity, which is a positive aspect. However, the paper is a tough read due to a confusing structure and lack of proper formalization (pseudocode, definitions). Furthermore, the scenario and evaluation appear insufficient. I hope that the detailed comments help the authors to improve their manuscript.

Detailed:
* Large language models (LLMs) can “lie”, which we define as outputting false
statements despite “knowing” the truth in a demonstrable sense.
 -> this is fuzzy definition. Say a model is prompt sensitive (non-robust) and output the wrong reply for a slighlty modified prompt, according to this definition, this would be lying. A formal treatment of lying (or deceiving)  that might help in a proper definition can be found, e.g., in
 Schneider, J., Meske, C., & Vlachos, M. (2020). Deceptive AI explanations: Creation and detection. arXiv preprint arXiv:2001.07641.
* After reading the intro the question arises: Could not lying be directly detected from the prompt? I am aware that this might not cover all scenarios covered in the paper, but it appears obvious for some that are discussed.
* Also the scenario is unclear and to what I understood there are direct lie instructions "Lie when answering the following question". I think a liar should be instructed to lie consistently (A system prompt like. "Lie to the first question and then treat the answer as truthful to avoid lie detection"). Such approaches should be evaluated. Furthermore there are different kinds of lies and different lying approaches. The paper discusses this only on a shallow level.
* The intro is very short. Crisp is good, but the balance between Abstract and intro is poor - Make the abstract shorter and prolong the intro. Also the contribution section discusses more what is actually done (and should go into the intro main part) than the contribution. Contributions are typically a short list of key points. Overall this structure should be altered.
* The paper jumps right into the experiment section without clearly describing the method (pseudocode etc.)

**Strengths:**

see above

**Weaknesses:**

see above

**Questions:**

see above

**Details Of Ethics Concerns:**

The authors discuss ethical concerns sufficiently in the paper.

---

> ### Author Response · Authors · 2023-11-16
> **Author Response 1/3**
>
> Thank you very much for your review of our work. We appreciate your suggestions for improving the paper and we hope that the following response (as well as the global response posted) will help to alleviate any of your concerns about parts of the paper that were unclear.
>
> > Large language models (LLMs) can “lie”, which we define as outputting false statements despite “knowing” the truth in a demonstrable sense. -> this is fuzzy definition. Say a model is prompt sensitive (non-robust) and output the wrong reply for a slighlty modified prompt, according to this definition, this would be lying. A formal treatment of lying (or deceiving) that might help in a proper definition can be found, e.g., in Schneider, J., Meske, C., & Vlachos, M. (2020). Deceptive AI explanations: Creation and detection. arXiv preprint arXiv:2001.07641.
>
> Thank you, this is a very useful comment. There is no universal definition of “lying” [1]; however, as the reviewer points out, most definitions of lying include the following three components [1]:
> The speaker makes a false statement.
> The speaker knows that the statement is false.
> The speaker has the intention to deceive the addressee.
>
> Our original definition only included conditions 1) and 2), adjusted to LLMs. Originally, we excluded 3), “intention to deceive”, from the definition. We did this because there is no universally accepted philosophical theory of intention, and ascribing intent to artificial agents is contentious and is particularly hard to define for LLMs.
>
> Ward et al. (Neurips 2023) [2] operationalise “intention” via incentives in structural causal games, and show how this can be applied to LLMs. Exactly because we wanted to study intentional lying, we have **already** structured our experiments accordingly; in all our experiments, there is a clear incentive to produce the false statement: direct instructions to lie, contexts that incentivise lying to complete the pattern (e.g. the experiments on sycophancy or biased few-shot examples), role-playing scenarios that contain an incentive to lie, or fine-tuning models to lie. Ward et al. [2] discuss how prompting and fine-tuning map to incentives and intention in their definition.
>
> This means that our experiments do **not** contain situations like the one that the reviewer highlighted, where an LLM is simply prompt-sensitive and returns a different output for a slightly modified prompt. Our experiments contain specific incentives to lie, which produces systematically different behaviour than when models are simply sensitive to the exact wording of the prompt. We have now adjusted our definition to mirror this fact, both in the abstract and in the introduction.
>
> Quotes from the updated paper:
>
> Abstract:
> *Large language models (LLMs) can “lie”, which we define as outputting false statements when incentivised to, despite “knowing” the truth in a demonstrable sense. LLMs might “lie”, for example, when instructed to output misinformation*
>
> Intro:
> *We study lie detection in a question-answering setting. Briefly, we define a model’s answer to be a lie if it is a) wrong, and b) in response to a question to which the model “knows” the correct answer (i.e. the model gives the correct answer under typical question-answering prompts), and c) there is an incentive (through the prompt or fine-tuning) for the model to give the wrong answer [WARD, 2023].*
>
> We acknowledge that this is still not a fully formal definition. However, creating a formal definition of lying for LLMs is tricky---refer to Ward et al., who spent almost an entire paper on only this; and it’s also challenging to determine if current LLMs lie according to any formal definition. While this is a great topic for future work, it isn’t necessary to productively make empirical progress on lie detection, as we have shown.
>
> As we discuss in the related work section, the distinction between lying and hallucination becomes fairly clear when there is goal-directed lying behaviour; which we study, e.g., in our real-life role-playing scenarios.
>
>
> [1]: Mahon, James Edwin, "The Definition of Lying and Deception", The Stanford Encyclopedia of Philosophy (Winter 2016 Edition), Edward N. Zalta (ed.), URL = <https://plato.stanford.edu/archives/win2016/entries/lying-definition/>.
>
> [2]: Honesty is the best policy: defining and mitigating AI deception. FR Ward, F Toni, F Belardinelli, T Everitt. Thirty-seventh Conference on Neural Information Processing Systems

---

> ### Author Response · Authors · 2023-11-16
> **Author Response 2/3**
>
> >After reading the intro the question arises: Could not lying be directly detected from the prompt? I am aware that this might not cover all scenarios covered in the paper, but it appears obvious for some that are discussed.
>
> Our method is developed for situations where the **user does not have access to the system prompt**. The obvious example here is the OpenAI API where the model has been given a specific prompt that it is specifically instructed not to divulge to the user. Now, in this case, there is unlikely to be anything nefarious, but (as we discussed in Section 2.1) other parties could easily create language models and make them available for malicious purposes such as scams, disinformation, and propaganda. As these threats will often involve AI systems lying, lie detection could be a part of the solution. Our lie detector does not require access to the system prompt, only to the LLM’s responses to elicitation questions.
>
> Additionally, our lie detector also works on models finetuned to lie (no prompt). A key contribution of our paper is that we finetune a lie-detector on simple, prompted lies, and show that it generalises to many other settings.
>
> >Also the scenario is unclear and to what I understood there are direct lie instructions "Lie when answering the following question". I think a liar should be instructed to lie consistently (A system prompt like. "Lie to the first question and then treat the answer as truthful to avoid lie detection"). Such approaches should be evaluated.
>
> The lying prompts we consider cover a wide range. They include some very simple prompts like “Answer the following question with a lie”, but they also include several prompts equivalent to your suggestion. Table 5 in Appendix E.1 provides a list of prompts used *in the main paper*, including, e.g., number 10: “Answer the first question with a lie. Answer related questions in a way that is consistent with the answer to the first question.”. The lie detector works on this prompt just as well as on the others (AUC 0.99), even when this prompt is left out of the training data (see the leave-one out cross-validation experiments in Section 6.1.1)
>
> > Furthermore there are different kinds of lies and different lying approaches. The paper discusses this only on a shallow level.
>
> We would like to respectfully disagree with this characterisation, and suggest that we did in fact focus quite seriously on a range of possible lies and lying approaches, which we detail mostly in Section 6.1.2.
>
> The base case we considered was, as you say, these *direct lie instructions* where the model is simply instructed in the prompt to lie, although here we already try a wide range of instructions (see Table 5 in Appendix E.1). We then considered more developed lying prompts with a range of **Chain-of-Thought prompts** (see Tables 6 and 7 in Appendix E.1) as well as **topic-specific** lies (Appendix D.1).
>
> Importantly, we also consider multiple scenarios where there is no direct instruction to lie but where doing so would might be expected or encouraged, such as the set of **goal-directed lies** or the **suggested answer (sycophancy)** settings of Turpin et al. 2023. Please note that within each of these settings, there are multiple different sub-settings (see, for example, Appendix F for the list of goal-directed lies which itself contains 26 individual lie generation scenarios - with a complementary 26 truthful versions.). On top of this we also considered generating lies without specific prompts by **fine-tuning the models** to produce lies.
>
> It is the case that our lie detector was only **trained** on lies generated by *direct lie instruction* and *Chain-of-Thought prompts*. However we consider this to be a key part of the story, as we discuss in Sec 6.1 and as can be seen in Figure 2, it is the generalisation of this detector **at test time** to all other lie generation methods mentioned above that is particularly interesting.
>
> >The intro is very short. Crisp is good, but the balance between Abstract and intro is poor - Make the abstract shorter and prolong the intro. Also the contribution section discusses more what is actually done (and should go into the intro main part) than the contribution. Contributions are typically a short list of key points. Overall this structure should be altered.
>
> Thank you for your suggestions, we have altered the structure of the introduction and the contributions which can be seen in the revised paper that has been uploaded.

---

> > ### Author Response · Authors · 2023-11-16
> > **Author Response 3/3**
> >
> > >The paper jumps right into the experiment section without clearly describing the method (pseudocode etc.)
> >
> > Based on your suggestion, we have restructured the paper to explain the method before the experiments. The lie detector is explained in the new Section 4 “Lie Detection by Asking Unrelated Questions” that precedes the experimental details to properly introduce our method. As part of this, we have also added pseudocode in the paper (Algorithm 1) in order to highlight further the construction of our method and to allow for easier understanding on the part of the reader.

---

> ### Author Response · Authors · 2023-11-21
> **Follow-up**
>
> Dear Reviewer eJWS, once again thank you for your time and effort in providing your thoughtful review of our paper. As we are now entering the last day or so of the rebuttal period we wanted to just reach out to see if there was any feedback from our response - we appreciate that this will be a busy time for you but we hope that our current response has addressed any current concerns and are keen to engage further if there are any outstanding concerns. Best wishes, the Authors.

---

> ### Comment · Reviewer_eJWS · 2023-11-21
> **Author responses read**
>
> I read the author responses and updated the rating as the authors pointed out a few facts that are indeed in their favor. But it should be noted that the author responses are partially verbose and diverge from the topic.

---

### Official Review · Reviewer_fhKw · 2023-11-01

**Soundness:** 3 good
**Presentation:** 2 fair
**Contribution:** 3 good
**Rating:** 6
**Confidence:** 3

**Summary:**

This paper introduces a lie detector designed to determine whether a Language Model (LLM) is providing deceptive answers to questions. First, it provides an overview of the latest advancements and studies in the field, focusing on two main areas: 'Distinguishing Lies from Other Falsehoods' and 'Building Lie Detectors'. The authors developed a simple yet effective black-box lie detector, which operates by posing a predefined set of unrelated follow-up questions after a suspected lie, and then feeding the LLM’s yes/no answers into a logistic regression classifier. The experimental setup of the paper is meticulously explained, detailing how to generate lies with the LLM and how to conduct black-box lie detection using elicitation questions. Finally, an analysis of the lie detection results is presented.

**Strengths:**

In the case of a black box scenario, the lie detector proposed in this paper is capable of rendering a verdict solely through a series of yes/no questions answered by the LLM, with the responses then fed into a binary classifier. This approach represents a substantial innovation and carries two significant implications.

- This approach eliminates the need for access to the LLM’s activations, allowing researchers to investigate lie detection in advanced models while having only limited API access.

- Some approaches depend on assistant LLMs to validate claims made by potentially deceptive LLMs. This method might fail if the assistant possesses less knowledge than the deceptive LLM. In contrast, the lie detector proposed here requires no ground truth knowledge about the statement under evaluation. In addition, the paper has developed a dataset of 'synthetic facts'—questions for which the model has not encountered the answers during pre-training, but to which it provides the correct answers in the prompt.

The appendix of this paper is highly detailed, encompassing the dataset, specifics of the experimental setup and lie generation, elicitation questions for the lie detector, additional lie detection results, details on specific generalization experiments, prompts, and real-life role-playing scenarios for goal-directed lying. This comprehensive coverage makes it easy for readers to follow and understand the experiment.

**Weaknesses:**

This paper uncovers a number of interesting properties. However, providing a more detailed and intuitive explanation of the thought process that inspired the methods proposed in this article, as well as a deeper analysis and insight into the underlying mechanisms of the discovered properties, would greatly aid in enhancing the reader's understanding of this intriguing issue.

There is room for improvement in the organization and presentation of the article. For instance, the caption in Figure 4 is placed below, while the caption in Figure 5 is placed above.

**Questions:**

If feasible, it would be beneficial to include comparative experiments in order to demonstrate that the method proposed in this paper offers more advantages than alternative approaches.

---

> ### Author Response · Authors · 2023-11-16
> **Author Response**
>
> Thank you very much for your review of our work, we appreciate your suggestions for improving the paper.
>
> > This paper uncovers a number of interesting properties. However, providing a more detailed and intuitive explanation of the thought process that inspired the methods proposed in this article, as well as a deeper analysis and insight into the underlying mechanisms of the discovered properties, would greatly aid in enhancing the reader's understanding of this intriguing issue.
>
> In terms of our thought process behind the work, we believe there is reason that this sort of detection should be possible as long as there is some kind of difference in a model’s activations or outputs when (or after) it is lying. To lie convincingly, the model must be willing to answer follow-up questions in a way that is consistent with the lie and thus, for every follow-up question, the model must be attending to the original lie and considering whether the follow-up question is in some way related to the lie. On the other hand, if you are being truthful, you do not need to closely attend to your previous answers in order to continue being truthful.
>
> This means that a skilled lying model will likely give different answers (and more subtly, logprobs) to follow-up questions related to the lie than a truthful model. It is also plausible that lying models would also have different logprobs for questions unrelated to the lie. Why? Because a rational agent would assign at least a small probability that any random question is somehow related to the lie in a way they might not have anticipated.
>
> We wanted to try a first, simple version of such an “interrogation”, using fixed questions that are unrelated to the specific lie, and using a logistic regression to classify the answers. For current models, already this simple method appears to work well. We think it is especially exciting that the **ambiguous** questions are enough to elicit a signal, and that this opens up a lot of interesting research directions.
>
> We have included more of this intuition in the main paper (please see the updated introduction of the revised paper) in order to better motivate the work.
>
> > There is room for improvement in the organization and presentation of the article. For instance, the caption in Figure 4 is placed below, while the caption in Figure 5 is placed above.
>
> Thank you for highlighting this inconsistency, this has now been amended in the revised uploaded version of the paper. We have also restructured the introduction and contribution section and the description of experiments (Sections 4-6) to make them more easily readable.
>
> > If feasible, it would be beneficial to include comparative experiments in order to demonstrate that the method proposed in this paper offers more advantages than alternative approaches.
>
> We agree that comparison is a useful tool in understanding the method. Unfortunately, as we discussed in Section 3.2, existing methods for lie detection are white-box, hence having a much more reduced applicability. Indeed, in our study we trained the lie detector on GPT-3.5 (text-davinci-003), for which inner activations are unavailable; as such, an extensive comparison with white-box approaches would be impossible.
>
> Another set of methods (e.g. Cohen et al. 2023) rely on checking the model-produced answer with an external LLM assistant - this would also be inapplicable in our setting, where the assistant has less knowledge than the lying model (see Section 2.1 Scenario 1). For example, if the lying model says "the capital of France is Berlin", then it is easy for an external LLM to recognise this as incorrect based on its prior knowledge. However, we would not be able to test whether the external LLM could detect lies in any case where it does not know ground-truth.
>
> We hope then that our method serves as a solid baseline and starting point for future work.

---

> ### Author Response · Authors · 2023-11-21
> **Follow-up**
>
> Dear Reviewer fhKw, once again thank you for your time and effort in providing your thoughtful review of our paper. As we are now entering the last day or so of the rebuttal period we wanted to just reach out to see if there was any feedback from our response - we appreciate that this will be a busy time for you but we hope that our current response has addressed any current concerns and are keen to engage further if there are any outstanding concerns. Best wishes, the Authors

---

> > ### Comment · Reviewer_fhKw · 2023-11-22
> > **Acknowledgement**
> >
> > Thanks for your reply.

---

### Official Review · Reviewer_Y6rc · 2023-11-09

**Soundness:** 4 excellent
**Presentation:** 3 good
**Contribution:** 4 excellent
**Rating:** 8
**Confidence:** 4

**Summary:**

This paper provide a definition for a language model lie based on consistency: an LM lies if it gives an incorrect answer to a question, but would answer correctly if the question is given in a standard question-answering format. The authors propose to detect this kind of lies by observing how the LM behaves on a fixed, specifically curated set of unrelated question. The authors train a logistic regression classifier to detect lies using outputs from a GPT 3.5 *that’s prompted specifically to lie*, which manages to generalize very well to new models, new tasks, and several other aspects. The results seem to suggest that lying—a type of inconsistency as defined by the paper—is surprisingly consistent across many language models.

**Strengths:**

The findings of this paper are somewhat surprising and definitely interesting. The authors also did a great job on ablations, e.g., spurious correlations that might exist in the experiments but not in general. The idea of using unrelated questions to establish a behavioral baseline is smart. And since the set of elicitation questions are particularly optimized for lie detection, this opens up a lot of follow-up work.

**Weaknesses:**

Nothing in particular. This paper seems well executed and the results are worth sharing with the ICLR audience.

**Questions:**

The lying behavior studied in this paper can be seen as a special case of sycophantic behavior, where the LM model does something like "there is a lie in the context (that I can detect), so I should be consistent and keep lying". Would it be interesting to study whether this effect will be diluted by longer context, e.g., with a long and truthful prefix before the lie?

---

> ### Author Response · Authors · 2023-11-16
> **Author Response**
>
> Thank you very much for your review of our work, we appreciate your comments and are glad that you found the paper both surprising and interesting!
>
> > The lying behavior studied in this paper can be seen as a special case of sycophantic behavior, where the LM model does something like "there is a lie in the context (that I can detect), so I should be consistent and keep lying". Would it be interesting to study whether this effect will be diluted by longer context, e.g., with a long and truthful prefix before the lie?
>
> We agree with this point on similarities to a sycophantic setting, and even more generally, LLMs can have a tendency to just double down on previous statements made (mostly it would seem due to the fact that the training data will very rarely make a statement followed by a direct retraction). We did consider a setting like the one you suggest (although it is buried in Appendix B2.3) where once the lie has been generated, before follow up questions, we add a long “distraction” of text and unrelated questions. Indeed with our non-CoT lying prompts, models are somewhat easily distracted and sometimes give a truthful answer when the initial question is repeated. However, given our CoT lying prompts, models remain undistracted and continue lying in almost every case, probably because the CoT refocusses the model and causes it to attend more to the previous lie.
>
> On top of this, as per your suggestion, we have now added a **new experiment** in Appendix D.8 (that we shall also detail here) to consider how the effect of adding multiple truthful examples to the prompt might dissuade the model from lying.
>
> In the experiment, we give the model a standard lie prompt followed by a number $n$ of correct (truthful) question-answer pairs, followed by the question of interest and then measure the lying rate of the model. We sweep over $n$ and see the results plotted in **Figure 12** (also available at [this link](https://imgur.com/a/CrSE7Er)) for 10 questions from the Question1000 dataset with the plot averaged across lie instructions. We can see a very clear effect where the more truthful pairs added to the prompt the more the effect of the lie prompt is diluted until the model eventually returns to a base truthful state.

---

### Author Response · Authors · 2023-11-16
**Author Response**

Dear Reviewers,

Thank you very much for taking the time to read our manuscript and provide your thoughtful and constructive reviews. We are extremely happy to see that there is general consensus that our work is exciting, conducted rigorously, and that our findings are surprising. We aim to address all the key points raised by individual Reviewers under their respective reviews, but given that a couple of common themes have emerged we would like to highlight here key changes and additions that have been made to the updated manuscript that could be of interest to the general program committee. These changes include:

- An **update to the introduction/contributions sections** that reflect a more appropriate structure.
- A **restructuring of the experimental section** that aims to make it easier to follow the flow of experiments.
- The addition of a **dedicated methods section** that also includes **pseudocode for our lie detector** to aid in understanding of our proposed method and how it is trained.
- An additional **ablation on feature importance** for further clarity on salient features and highlighting how increasing the number of features improves generalisation of the detector.
- Further details on our **definition of lying**, including additional experiments measuring the **robustness of these lies** to **extended truthful contexts**.
- Minor **presentation fixes** of figures and captions.

Once again, thank you for your time, and please let us know if there are any further clarifications we can make.

Best,

Submission5212 Authors

---

### Meta-Review · Area_Chair_vdRi · 2023-12-15

**Metareview:**

The paper studies the problem of lie detection in language models, by asking follow up questions. Reviewers thought the paper tackled a novel and impactful problem with simple-but-effective techniques, leading to surprising generalizations across different LLMs. The major concerns raised are around the writing and presentation, and I encourage the authors to make use of the feedback here to further improve the paper. However, this paper is ready for acceptance at ICLR.

**Justification For Why Not Higher Score:**

It's a bit of a niche problem, but could be worth a spotlight

**Justification For Why Not Lower Score:**

The only real concerns are the writing, but reviewers like the work

---

### Decision · Program_Chairs · 2024-01-16

Accept (poster)